# Revealing interpretable object representations from human behavior

**Charles Y. Zheng**
Section on Functional Imaging Methods
National Institute of Mental Health
charles.zheng@nih.gov

**Francisco Pereira**
Section on Functional Imaging Methods
National Institute of Mental Health
francisco.pereira@nih.gov

**Chris I. Baker**
Section on Learning and Plasticity
National Institute of Mental Health
martin.hebart@nih.gov

**Martin N. Hebart**
Section on Learning and Plasticity
National Institute of Mental Health
martin.hebart@nih.gov

## Abstract

To study how mental object representations are related to behavior, we estimated sparse, non-negative representations of objects using human behavioral judgments on images representative of 1,854 object categories. These representations predicted a latent similarity structure between objects, which captured most of the explainable variance in human behavioral judgments. Individual dimensions in the low-dimensional embedding were found to be highly reproducible and interpretable as conveying degrees of taxonomic membership, functionality, and perceptual attributes. We further demonstrated the predictive power of the embeddings for explaining other forms of human behavior, including categorization, typicality judgments, and feature ratings, suggesting that the dimensions reflect human conceptual representations of objects beyond the specific task.

## 1 Introduction

A central goal in understanding the human mind is to determine how object concepts are represented and how they relate to human behavior. Given the near-infinite number of tasks or contexts of usage, this might appear to be an impossible prospect. Take the example of picking a tomato in a grocery store. A person may recognize it by its color, shape, size, or texture; they may also know many of the more conceptual dimensions of a tomato, such as it being a fruit, or functional ones such as being a salad item. In most cases, however, only a few of these aspects might matter, depending on task context (e.g. as a grocery item or as a projectile in an audience that is not pleased with a presentation). Thus, to understand how we interact meaningfully with the objects around us, we need to tackle three problems simultaneously: (1) determine the object concept, the cognitive representation of behaviorally-relevant dimensions that distinguish an object from other objects. (2) determine which object concept representations are integrated into our perceptual decisions. (3) characterize the influence of the context – determined by the surrounding objects – on those decisions.

There have been many attempts to represent object concepts in terms of *semantic features*, a vector of variables indicating the presence of different aspects of their meaning. These representations have been used to model phenomena such as judgments of typicality or similarity between concepts, or reaction times in various semantic tasks, with the goal of drawing conclusions about mental representations of those concepts (see Murphy (2004) for an extensive review of this literature). These features have usually been binary properties, postulated by researchers. A landmark study McRae et al. (2005) departed from this approach by instead asking hundreds of subjects to name binary properties of 541 objects, yielding 2,526 such semantic features. These corresponded to different types of information, e.g. for the concept "hammer" subjects might list taxonomic ("is a tool"), functional ("is used to hammer nails"), perceptual ("heavy"), among others. The results revealed that concepts for objects in the same basic level category shared many features; at the same

time, there were also features that distinguished even very similar concepts. This effort was later replicated and extended by Devereux et al. (2014), generating 5,929 semantic features for 638 objects.

The main issues with features produced in either study are the lack of degree (they can only be present or absent, albeit with varying naming frequencies), the extreme specificity of some features, and the omission of many features shared by the majority of concepts. A separate concern is the fact that, absent a specific context of use for each object, subjects will not likely think of many valid properties (e.g. that a tomato can be thrown). A different approach is to postulate the existence of certain semantic features and ask subjects to judge the degree to which features are salient for each concept, instead of assuming binary features. Binder et al. (2016) did this for 65 features corresponding to types of information for which there is evidence of brain representation (in their terminology: sensory, motor, spatial, temporal, affective, social, cognitive, etc). This requires experts to specify the features in advance, and not all features can be judged easily by salience. In all three approaches outlined above, there is also no clear way of determining which features are critical to semantic behavior.

Here we introduce an approach that uses *only* information from behavioral judgments about the grouping of object images *in the context of other objects* to estimate representations of object concepts. We demonstrate that this approach can predict human behavior in face of new combinations of objects and that it also allows prediction of results in other behavioral tasks. We show that individual dimensions in the low-dimensional embedding represent complex combinations of the information in the binary features in Devereux et al. (2014) and, furthermore, are interpretable as conveying taxonomic, functional, or perceptual information. Finally, we will discuss the way in which this representation suggests a simple, effective model of judgments of semantic similarity in context.

## 2 DATA AND METHODS

### 2.1 THE ODD-ONE-OUT DATASET

The collection of the Odd-one-out behavioral dataset is an ongoing project by the authors. The dataset contains information about 1,854 different object concepts, which we will denote by $c_1, \ldots, c_m$ (e.g. $c_1$ = 'aardvark',..., $c_{1854}$ = 'zucchini'). A triplet consists of a set of three concepts $\{c_{i_1}, c_{i_2}, c_{i_3}\}$, for instance, $\{c_2, c_{117}, c_{136}\}$ = {'abacus', 'beetle', 'bison'}, presented to Amazon Mechanical Turk (AMT) workers as three photographs of exemplars, one for each of the three concepts. Importantly, these photographs were accompanied neither by labels or captions, meaning behavior was based only on those pictures. Workers were asked to consider the three pairs within the triplet $\{(c_{i_1}, c_{i_2}), (c_{i_1}, c_{i_3}), (c_{i_2}, c_{i_3})\}$, and to decide which item had the smallest similarity to the other two (the "odd-one-out"). This is equivalent to choosing the pair with the greatest similarity. Let $(y_1, y_2)$ denote the indices in this pair, e.g. for 'beetle' and 'bison' they would be $(y_1, y_2) = (117, 136)$.

The large number of objects in this dataset has two different motivations. The first is obtaining behavioral data about objects from a wide range of semantic categories (tools, animals, insects, weapons, food, body parts, etc), rather than the much smaller numbers in prior studies. We will use the term "category" loosely, to refer to a taxonomic level above that of the individual object; this is often, but not always, a basic level category (Murphy, 2004). The second is to have subjects consider each object in as many different contexts as possible. We take the contexts instantiated by comparing each object to the others in the triplet as a proxy of the many situations each object might appear or be used in. For instance, "tomato" might be paired with "milk" if the third object in the triplet were "table" ("tomato" and "milk" are food), but perhaps not if the third object was "wine" ("milk" and "wine" are a more specific kind of food). If "tomato" were presented with "lipstick" and "rock", it could be paired with the former (because they are red) or the latter (because they can both be projectiles). Finally, the presence of several objects sharing a given semantic category means that there is redundancy. Hence, one should be able to predict behavior in face of an object in novel contexts, if similar objects have already been observed in those contexts.

The dataset in this paper contains judgments on 1,450,119 randomly selected triplets, roughly $0.13\%$ out of all possible triplets using the 1,854 concepts. These were the triplets remaining after excluding AMT participants that showed evidence of inappropriate subject behavior, namely if responses were unusually fast, exhibited systematic response bias, or were empty (137,281 triplets or $8.65\%$). We plan to collect additional triplets and release all data by the end of the study.

## 2.2 Sparse POsitive Similarity Embedding (SPoSE)

**Semantic representation of concepts**    Our method, Sparse Positive Similarity Embedding (SPoSE) is based on three assumptions. The first is that each object concept $c_i$ can be represented by semantic features, quantified in a vector $\mathbf{x_i} = (x_{i1}, \ldots, x_{ip})$. These features represent aspects of meaning of the concept known by subjects, and correspond to dimensions of a space where the geometrical relationships between vectors encode the (dis)similarity of the corresponding concepts. We estimate semantic features *from behavioral data alone*, using the probabilistic model described in the next section. In our model, each feature/dimension $x_{if}$ in the vector $\mathbf{x_i}$ is real and non-negative, so as to make it interpretable as the *degree* to which the aspect of meaning it represents is present and influences subject behavior (Murphy et al., 2012). Further, we expect features/dimensions to be sparse McRae et al. (2005), which led us to add a sparsity parameter to our model.

Based on related work in sparse positive word embeddings of words from text corpora (Murphy et al., 2012), we expect some of the dimensions to indicate membership of the semantic category the object belongs to (e.g. tool, vegetable, vehicle, etc). If two objects in a triplet share a category and the third does not, subjects will tend to group objects with a common category with very high probability. However, categories alone cannot fully explain subject performance in this task, as there may be triplets where either (a) all three objects belong to the same category or (b) no two objects share a category. We expect our method to find non-taxonomic information similar to that in McRae et al. (2005) and Devereux et al. (2014), if it allow us to explain semantic decision making behavior.

**Probabilistic model**    Our second assumption is that the decision in a given trial is explained as a function of the similarity between the embedding vectors of the three concepts presented. Given two concepts $c_i$ and $c_j$, the similarity $S_{ij}$ is the dot product of the corresponding vectors $\mathbf{x_i}$ and $\mathbf{x_j}$:

$$S_{ij} = \langle \mathbf{x_i}, \mathbf{x_j} \rangle = \sum_{f=1}^{p} x_{if} x_{jf}. \tag{1}$$

Our third assumption is that the outcome of the triplet task is stochastic. This assumption reflects both trial-to-trial inconsistencies in how an individual subject makes the decision, as well as subject differences in the decision-making process. Given that we pool data across many subjects, we do not distinguish between these two sources of randomness. We model the probability that a subject will choose a given pair out of the three possible pairs as proportional to the exponential of the similarity,

$$\Pr[y_1, y_2] \propto \exp(S_{y_1, y_2}).$$

Since probabilities of the three possible outcomes add to one, this gives

$$\Pr[y_1, y_2] = \frac{\exp(S_{y_1, y_2})}{\exp(S_{i_1, i_2}) + \exp(S_{i_1, i_3}) + \exp(S_{i_2, i_3})}. \tag{2}$$

The vectors $\mathbf{x_i}$ in this model are not normalized, in that each dimension – and, therefore, vector similarity – can be arbitrarily large. In addition to the reasons described in the previous section, this allows us to model situations where the decision is close to being deterministic (e.g. two concepts are so similar that they will always be grouped together, regardless of third concept). We also tried modeling choice probabilities using Euclidean distances (rather than dot product) between embedding vectors, but this resulted in similar (but overall slightly worse) results on our evaluations.

**Parameter fitting**    We infer embedding vectors $(\mathbf{x_1}, \ldots, \mathbf{x_m})$ from data by fitting a regularized maximum likelihood objective function. Let $n$ be the total number of triplets available for building the model. For the $j^{th}$ triplet in the dataset, let $(i_{1,j}, i_{2,j}, i_{3,j})$ denote the indices of the concepts in the triplet, and let $(y_{1,j}, y_{2,j})$ indicate the pair of concepts chosen, for triplet $j = 1, \ldots, n$. We randomly split the dataset into $n_{train}$ triplets for training and $n_{val}$ for choosing the sparsity regularization parameter $\lambda$. We index the training set by $j = 1, \ldots, n_{train}$ and the validation set by $j = n_{train} + 1, \ldots, n$.

The objective function for SPoSE, given a specified number of dimensions $p$, is

$$\sum_{j=1}^{n_{train}} \log \left( \frac{\exp(\mathbf{x_{y_{1,j}}}^T \mathbf{x_{y_{2,j}}})}{\exp(\mathbf{x_{i_{1,j}}}^T \mathbf{x_{i_{2,j}}}) + \exp(\mathbf{x_{i_{1,j}}}^T \mathbf{x_{i_{3,j}}}) + \exp(\mathbf{x_{i_{2,j}}}^T \mathbf{x_{i_{3,j}}})} \right) + \lambda \sum_{i=1}^{m} ||\mathbf{x_i}||_1 \tag{3}$$

Here $|| \cdot ||_1$ is the L1 norm, so $||\mathbf{x}||_1 = \sum_{f=1}^{p} |x_f|$, and $x_f \geq 0$ for $f = 1, \ldots, p$.

We use the Adam algorithm (Kingma & Ba, 2015) with an initial learning rate of 0.001 to minimize the objective function, using a fixed number of 1,000 epochs over the training set, which was sufficient to ensure convergence. Since this objective function is non-convex, Adam is likely to find only a local minimum. However, the combination of non-negativity and L1 penalty leads to very similar solutions across random initializations; we discuss this further in Section 3.1.

We select the regularization parameter $\lambda$ out of a grid of candidate values by choosing the one that achieves the lowest cross-entropy $\mathrm{CE}_v$ on the validation set, defined by

$$\mathrm{CE}_v = \sum_{j=n_{train}+1}^{n} \log \left( \frac{\exp(\mathbf{x_{y_1,j}}^T \mathbf{x_{y_2,j}})}{\exp(\mathbf{x_{i_1,j}}^T \mathbf{x_{i_2,j}}) + \exp(\mathbf{x_{i_1,j}}^T \mathbf{x_{i_3,j}}) + \exp(\mathbf{x_{i_2,j}}^T \mathbf{x_{i_3,j}})} \right).$$

The dimensionality of the embedding, $p$, can be determined heuristically from the data. If $p$ is set to be larger than the number of dimensions supported by the data, the SPoSE algorithm will shrink entire dimensions towards zero, where they can be easily thresholded. We believe this happens because the L1 penalty objective encourages redundant similar dimensions to be merged, which would not happen with an L2 penalty; we provide a more formal justification in the Supplementary Material.

## 2.3 RELATED WORK

There are several existing methods for estimating embeddings of items from behavioral judgments. Agarwal et al. (2007) introduced the *generalized non-metric multidimensional scaling* (GNMMDS) framework as a generalization of non-metric MDS. The behavioral task they study is a 'quadruplet' task, where a subject is shown two pairs of items, $c_i, c_j$ and $c_k, c_\ell$. Then the subject is asked to decide if the similarity between $c_i$ and $c_j$ is greater or less than the similarity between $c_k$ and $c_\ell$. GNMMDS learns embedding vectors for the concepts, so that the Euclidean distance between embedding vectors approximates the dissimilarity between the respective items. In the two-alternative forced-choice (2AFC) task of Xu et al. (2011), the subject is shown an anchor $c_i$ and two alternatives $c_i, c_j$. Xu et al. (2011) study this task, under the assumption one already has parametric features $z_i$ for the concepts $c_i$. In the triplet task of Tamuz et al. (2011), the subject is shown an anchor $c_i$ and asked which of $c_j$ or $c_k$ is more similar to $c_i$. This is also a special case of the quadruplet task, with pairs $(c_i, c_j)$ and $(c_i, c_k)$. Tamuz et al. (2011) use a model where $\Pr[(c_i, c_j) \text{ more similar than } (c_i, c_k)] = \frac{\mu + d_{ik}^2}{2\mu + d_{ij}^2 + d_{ik}^2}$, where $d_{i,j} = ||x_i - x_j||^2$ and where $\mu$ is some constant to be determined empirically.

Similarly, Wah et al. (2014) show a series of adaptive displays for an anchor $c_i$, where the subject must partition the queries $c_j, c_\ell, \ldots$ into a set of similar and a set of dissimilar queries. In contrast to our work, the aforementioned studies did not use sparsity or positivity constraints, nor did they intend to evaluate the interpretability of the embedding.

Our requirements of having a sparse, positive vector representing each concept, and a three-choice probabilistic model, led us to develop a new method. It is not straightforward to extend the probabilistic model of Xu et al. (2011) to a task with more than 2 choices; it also requires an *a priori* feature matrix for the concepts, which is not available for our dataset. GNMMDS could potentially be applied to our data with added sparsity and positivity constraints, but does not produce probability estimates for the task. Finally, the model in Tamuz et al. (2011) could be extended to the three-choice triplet task with sparsity and positivity constraints, but it would require an extra tuning parameter $\mu$, in addition to the L1 sparsity penalty $\lambda$ which is needed to obtain sparse embeddings.

An alternative approach for representing concepts is the use of word vector embeddings, derived from information about co-occurrence of words in very large text corpora, *instead* of behavior. Pilehvar & Collier (2016) introduced a method for estimating vectors for each of the different concept senses (synsets) of a particular word, from its word2vec (Mikolov et al., 2013) embedding vector and the synset relationships in the WordNet ontology (Miller, 1995). Each of our concepts has a corresponding synset and, therefore, can be represented by a synset vector. However, the latter are real-valued and dense, and hence do not satisfy any of our requirements. There is an embedding method that enforces positive, sparse values for each dimension (NNSE, Murphy et al. (2012)), but it produces one vector per word rather than synset. Both methods have been used to generate behavioral predictions. Hence, we will use them as an alternative concept representation in our experiments.

## 3 EXPERIMENTS

### 3.1 DATA AND EXPERIMENTAL SETUP

We used Tensorflow (Abadi et al., 2015) to fit the model (3) to the 1,450,119 triplets collected, using a 90-10 train-validation split to pick the regularization parameter $\lambda$. Searching over parameters $\{0.0070, 0.0072, \ldots, 0.0100\}$ we identified the parameter $\lambda = 0.008$ with the lowest loss on the validation set. We initialized the model with 90 dimensions. After convergence, many of the dimensions were very small and very sparse, with an average value of less than 0.02 per item; the most dense dimension had maximum value of 2.3 and average value of 0.64. We discarded the small dimensions, yielding the 49-dimensional embedding used in all experiments. The number of non-zero dimensions varied across object categories, with a median of 14 (minimum: 3, maximum: 29).

Due to random initialization and batch ordering, the solution will vary randomly corresponding to different local minima. To check the reproducibility of the embedding we found, we ran the model 11 additional times with the same parameters but different random seeds. The number of resulting dimensions varied from 47 to 50, and between any two model fits, an average of 38.75 dimensions could be matched between the two models with a correlation of 0.8 or higher.

For comparison with vectors derived from text information, we used synset Pilehvar & Collier (2016) and NNSE Murphy et al. (2012)) vectors. We used 50-D vectors, corresponding to the dimensionality of our embedding, as well as 300-D (synset) and 2500-D (NNSE), corresponding to the best performing embeddings according to the original publications.

### 3.2 GENERALIZATION ON THE SAME TASK AND SIMILARITY PREDICTION

**Accuracy at predicting triplet decisions** In order to test our model, we collected an independent test set of 25,000 triplets, with 25 repeats for each of 1,000 *randomly selected* triplets; none of these were present in the data used to build the model. After applying the same quality control, there remained 614 unique triplets with at least 20 repeats. Having this many repeats allows us to be confident of the probability of response for each triplet. Furthermore, it allows us to establish a model-free estimate of the Bayes accuracy, the best possible accuracy achievable by any model. Since the optimal classifier predicts the population majority outcome for any triplet, this accuracy ceiling is therefore the average probability of the majority outcome over the set of all triplets (ties broken randomly). The ceiling estimated in this fashion was 0.673. The accuracy of our model was 0.637, above all baselines (see Table 1). Note that, as only 1,097 of our 1,854 objects are in the NNSE lexicon, the evaluation set for NNSE results was reduced to 129 unique triplets.

Table 1: Key results on prediction tasks, using SPoSE, synset, and NNSE embedding vectors.

| | $SPoSE_{49}$ | $Synset_{50}$ | $Synset_{300}$ | $NNSE_{50}$ | $NNSE_{2500}$ |
|---|---|---|---|---|---|
| Accuracy on 1,854 object test set | 0.637 | 0.486 | 0.493 | 0.397 | 0.491 |
| Accuracy on 48 object test set | 0.592 | 0.400 | 0.425 | 0.331 | 0.412 |
| Similarity on 48 objects ($R$) | 0.899 | 0.295 | 0.389 | 0.005 | 0.256 |
| CSLB prediction (median AUC) | 0.897 | 0.873 | 0.897 | 0.718 | 0.569 |
| Typicality (median $R$) | 0.545 | 0.418 | 0.520 | -0.254 | 0.555 |
| Category prediction (23-way accuracy) | 0.846 | 0.713 | 0.846 | 0.336 | 0.786 |

**Prediction of similarities** In addition to prediction accuracy, we wanted to test ability of the model to capture fine semantic distinctions, even though it was trained on extremely sparse, noisy data. To this effect, we collected an independent test set containing *every possible triplet* for 48 objects chosen out of the 1,854, one representative of each of 48 common categories (see list in Supplementary Material). This dataset contained 43,200 triplets after quality control; there were $17,296 = \binom{48}{3}$ unique triplets, with two or three subject answers for each. For this particular evaluation we re-trained our model on a dataset excluding all the triplets containing any 2 of the 48 objects. We then used the probability model (2) to predict the choice with the highest probability for each triplet in the 48-object dataset. This experimental setup ensured that any generalization had to rely on the observed relationships between objects *outside* the 48. It was also significantly harder than a random set of

triplets, given that no two concepts in any triplet were in similar semantic categories. We obtained an accuracy of $0.592$, above all the baseline methods (see Table 1).

A different way of examining prediction quality is to pool data across many triplets, by considering similarities between *pairs* of objects. The pairwise similarity for objects $A$ and $B$, $S_{AB}$, is obtained by dividing the number of triplets where subjects group $A$ and $B$ by the total number of triplets where $A$ and $B$ both appear. We calculated this similarity matrix $S$ for the 48-object dataset, as displayed in Figure 1 left. We then computed the expected similarity matrix according to our model. The Pearson correlation coefficient between off-diagonal entries of our prediction and the pairwise similarity derived from the 48-concept dataset was $0.899$, above all the baseline methods (see Table 1). This is particularly important because it demonstrates that sparse sampling of triplet data still allows accurate prediction of similarity relations between objects.

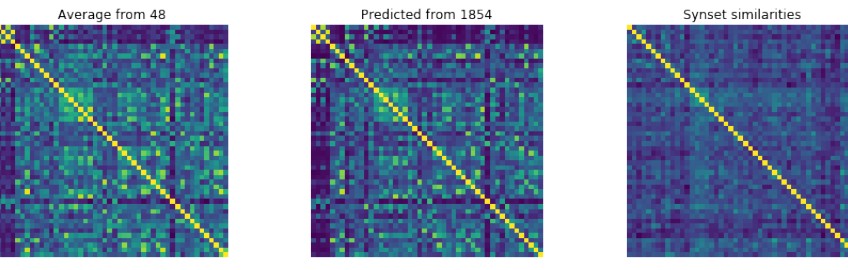

Figure 1: **Left:** the $48 \times 48$ matrix of similarities from the 48-concept dataset, defined as the pairwise probability of $A$, $B$ being matched in a triplet with a random third concept from the 46 remaining. **Center:** predicted similarity matrix using SPoSE vectors. **Right:** same, with synset vectors.

### 3.3 PREDICTION OF HUMAN-GENERATED SEMANTIC FEATURES

The second evaluation of our semantic vector representation was to test whether it could be used to predict the features in human-generated semantic feature sets: McRae (McRae et al., 2005) and CSLB (Devereux et al., 2014). CSLB is more extensive than McRae, both in the number of features (5,929) and objects (638), and shares 413 objects in common with the intersection of the objects from our 1,854 object set and the NNSE lexicon; therefore, we limited our evaluation to CSLB. Since many features are specific to only a few objects, they are sparse or blank when restricted to the 413 objects. Hence, we selected the 100 features with the highest density as prediction targets; the lowest density feature appeared in 23 of the 413 objects. We fit a separate L2-regularized logistic regression to predict each of the 100 features from SPoSE or baseline vectors. The prediction was done using 10-fold cross-validation, with a nested 10-fold cross-validation within each training set to pick the value of the regularization parameter. We evaluated the prediction of each CSLB feature by computing the area under the curve (AUC) as the bias term was varied, reported on Table 1.

These results demonstrate that SPoSE dimensions contain the information present in the main CSLB features for the 413 objects considered. This is also the case for 300-D synset vectors, but using almost 10 times as many dimensions as SPoSE. Although correlation between SPoSE and synset vector AUCs was $0.953$, SPoSE AUC was significantly higher in a few dimensions. These are related to shapes and other perceptual aspects ("is pink", "is small", "is flat","is white"); we hypothesize that this is due to their having a strong effect on subject behavior, but being less salient in text corpora.

### 3.4 INTERPRETABILITY OF LEARNED SEMANTIC DIMENSIONS

The goal of this analysis was to show that SPoSE dimensions can be "explained" in terms of the elementary CSLB features, by testing how well the former predicted the latter. We fit a L1-regularized non-negative regression model (NNLS) to predict each of the 49 SPoSE features from the 5,929 CSLB features, for all 496 concepts present in both datasets. We used a 10-fold cross-validation, with a nested 10-fold cross-validation within each training set to pick the value of the regularization parameter. The median correlation between each feature and its prediction was $0.58$, indicating that most SPoSE features can be predicted well. We then fit a model to the entire dataset, in order to use the NNLS regression weights to determine which CSLB features most influenced the prediction

of each SPoSE dimension. Figure 2 displays the most influential CSLB features for four different SPoSE features, selected to highlight the different types of information extracted.

Consider first Dimension A. The corresponding panel shows pictures of the 4 objects with the highest weights for this SPoSE dimension. To the right of the pictures, we list the 12 most important CSLB features for predicting the dimension. Their respective NNLS weights are shown by the size of the corresponding bars, and the CSLB feature type by the color (e.g. green is taxonomic). Dimension A appears to indicate the degree to which an object belongs to the "animal" semantic category. There appear to be multiple such category indicator dimensions, e.g. "food", "clothes", "furniture" (shown in the Supplementary Material). Consider now Dimensions B and C. These are examples of dimensions that correspond strongly to the type of material in objects, and that are well explained by CSLB features such as "made of metal" or "made of wood". These can *also* reflect category membership, if they are common characteristics of members of a given category. Finally, consider Dimension D. This is an example of a dimension that captures purely visual aspects, e.g. it is very strongly explained by the "is red" CSLB feature, together with other visual/perceptual ones. See Supplementary Material for a similar visualization for all 49 SPoSE dimensions.

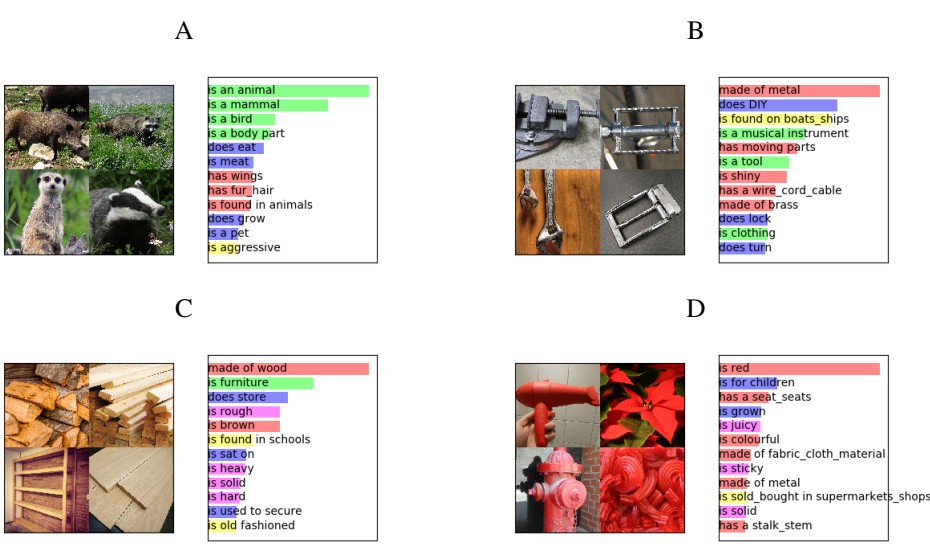

Figure 2: SPoSE dimensions explained by CSLB labels. The top 4 objects for each SPoSE dimension are shown with the 12 CSLB features with the largest weights in the NNLS model for predicting that feature. Bar length indicates relative weight, while color indicates feature type: green (taxonomic), blue (functional), yellow (encyclopedic), red (visual perceptual), violet (non-visual perceptual).

## 3.5 PREDICTION OF OTHER BEHAVIORAL OR HUMAN-ANNOTATED DATA

**Prediction of typicality ratings** The typicality of an object with respect to its semantic category is a graded notion of category membership (e.g. "robin" is a more typical bird than "chicken"). As we saw earlier, concepts for objects in the same semantic category tend to share SPoSE dimensions. Hence, a natural test is to assess the degree to which distances between semantic vectors for objects within a category reflect their typicality. To this effect, we collated a variety of pre-existing norms of typicality or naming frequency (Rosch (1975),McCloskey & Glucksberg (1978),Gruenenfelder (1984),Van Overschelde et al. (2004)) and used a collaborative filtering method (Koren, 2008) to extract a single aggregate norm from these for as many objects as possible. After this, we matched objects from the aggregate typicality norms to our 1,854 items and NNSE lexicon. Given that correlation estimates can be noisy for small samples, we restricted the analysis to the 9 categories with 25 or more objects in our set. For a given category, we computed a SPoSE vector for each category centroid, the mean of the vectors for all objects within that category. Next, we computed a proxy for the typicality of each object by calculating the dot product between its vector and the centroid vector. Finally, we calculated the correlation between the proxy typicality scores and the typicality scores from the norms. The median correlation with typicality across categories was 0.55.

After correcting for multiple comparisons, 5 of these results were statistically significant (weapon, vehicle, clothing, vegetable, fruit, respectively, 0.69, 0.67, 0.66, 0.62, 0.55), whereas the other 4 were not (furniture, body part, tool, animal, respectively 0.43, 0.42, 0.34, -0.14). Results are comparable for 300-D synset vectors and 2500-D NNSE vectors, and above the other baselines (see Table 1). These results suggest that SPoSE vectors for more typical objects are more similar to each other than for atypical ones. In combination with the fact that vectors are sparse, this suggests that more categorical SPoSE features also reflect its degree.

**Semantic category clustering**   Our final evaluation focused on determining the degree to which objects in the same semantic category cluster compactly in the embedding space. We ran a classification task for assigning 318 objects to one of 23 categories with $\geq 5$ items, as defined in Battig & Montague (1969). Leaving out each object in turn, we assign it the category of its nearest neighbor by cosine similarity; this is close to the task in Murphy et al. (2012), but using 23-way accuracy rather than cluster purity. The resulting accuracies were 0.846 for both SPoSE and 300-dimensional synset, and above all other baselines (see Table 1).

## 4   DISCUSSION AND CONCLUSIONS

In this paper, we show that human behavioral judgments are well-explained by a strikingly low-dimensional semantic representation of concrete concepts. This representation, which embeds each object in a 49-dimensional vector, allows prediction of subject behavior in face of new combinations of concepts not encountered before, as well as prediction of other behavioral or human-annotated data, such as typicality ratings or similarity judgments. Moreover, the representation is readily interpretable, as positive, sparse dimensions make it easy to identify which concepts load on each dimension. Further, we demonstrate that the value of each dimension in this space can be explained in terms of elementary features elicited directly from human subjects in publicly available norms. Given this converging evidence, we conclude that dimensions represent distinct types of information, from taxonomic (indicators of category membership) to functional or perceptual.

As the representations were estimated solely from behavioral data, this suggests a simple model of decision making in the triplet task. This can be viewed in terms of the distinction discussed in Navarro & Lee (2004) for judging concept similarity from semantic feature vectors. They distinguish a *dimensional* approach for representing stimuli (each feature is a continuous value, each concept is a point in a high-dimensional space, and similarity corresponds to proximity in the space), and a *featural* approach (each feature is binary, or discrete, and similarity is a function of the number of features that are common to both concepts, or that distinguish them). More refined schemes use modified distance metrics (dimensional) or combine commonality and distinctiveness (featural).

The use of sparsity and positivity in the SPoSE representation, and the vector dot product for computing concept similarity, blends the featural and dimensional approaches when making decisions about a triplet of concepts. First, if any two concepts share a semantic category, and the other one does not, the two concepts will likely be grouped together. Because of sparsity, the dot product between concepts will be driven primarily by the *number* of features are shared between the two concepts in the same category, versus the different one. Second, if any three concepts share a semantic category, they also share most, if not all of their non-zero features. The decision becomes a function of the *values* of the features shared between them, and hence dimensional rather than featural. Third, if all concepts belong to different categories, there may be very few features in common between any two of them. The results will likely be determined by which of those few features takes a higher value. Results might be idiosyncratic, e.g. two objects grouped because their pictures are both very red, while the alternative grouping would be because they are both string-like, and the former feature is more salient. This is another reason why our features are unbounded: their scale can reflect their importance in decision making. This is akin to learning a distance metric in dimensional approaches.

Our object representations capture the information that is necessary to explain subject behavior in the triplet task. Obviously, subjects have a lot more information about each concept that is not necessary or relevant for task performance. A promising direction for further work is to sample additional triplets so as to obtain more fine-grained, within-category distinctions. Beyond this, we have also considered the possibility of there being information influencing behavior that might be too infrequent to be estimated from this type of data, or elicited from human subjects. Yet another possible extension

is to consider different types of similarity judgments (Veit et al., 2017), e.g. resulting from asking subjects to group objects based on a specific attribute (size, color, etc.). One avenue for trying to identify this type of information is to predict synset vectors from SPoSE vectors and/or semantic features elicited from subjects, and represent the residuals in terms of a dictionary of new sparse, positive concept features. These could then be used as a complement to SPoSE dimensions.

## ACKNOWLEDGEMENTS

This research was possible thanks to the support of the National Institute of Mental Health Intramural Research Program (ZIA-MH-002909, ZIC-MH002968) and a Feodor-Lynen fellowship of the Humboldt foundation awarded to MNH. Portions of this study used the high-performance computational capabilities of the Biowulf Linux cluster at the National Institutes of Health, Bethesda, MD (biowulf.nih.gov). We would like to thank Alex Martin and Patrick McClure for their feedback and suggestions.

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

APPENDIX

| Word | Synset |
|------|--------|
| pizza | pizza.n.1 |
| beaver | beaver.n.6 |
| macadamia | macadamia_nut.n.2 |
| tarantula | tarantula.n.2 |
| clothes | clothes.n.1 |
| doormat | doormat.n.2 |
| hammer | hammer.n.2 |
| mistletoe | mistletoe.n.2 |
| washcloth | washcloth.n.1 |
| drain | drain.n.3 |
| furnace | furnace.n.1 |
| tea | tea.n.1 |
| chariot | chariot.n.2 |
| coaster | coaster.n.3 |
| music box | music_box.n.1 |
| rolling pin | rolling_pin.n.1 |
| knob | knob.n.2 |
| bookshelf | bookshelf.n.1 |
| candelabra | candelabra.n.1 |
| table | table.n.2 |
| metal detector | metal_detector.n.1 |
| bumper | bumper.n.2 |
| turban | turban.n.1 |
| flagpole | flagpole.n.2 |
| plaster cast | plaster_cast.n.1 |
| tuba | tuba.n.1 |
| camera | camera.n.1 |
| rudder | rudder.n.2 |
| canvas | canvas.n.2 |
| dice | dice.n.1 |
| toolbox | toolbox.n.1 |
| trigger | trigger.n.1 |
| bowler hat | bowler.n.3 |
| headphones | headphone.n.1 |
| file | file.n.4 |
| bench | bench.n.1 |
| navel | navel.n.1 |
| canister | canister.n.2 |
| tiara | tiara.n.1 |
| hopscotch | hopscotch.n.1 |
| trophy | trophy.n.2 |
| punching bag | punching_bag.n.2 |
| jet | jet.n.1 |
| telegraph | telegraph.n.1 |
| bag | bag.n.1 |
| laptop | laptop.n.1 |
| tape measure | tape.n.4 |
| bucket | bucket.n.1 |

Figure 3: The 48 concepts used to create the densely sampled test dataset.

## SPARSITY PROPERTIES

Empirically, we find that when the model is initialized with too many dimensions, SPoSE does not use all of the dimensions: it leaves many of them to be close to zero. (The reason they are not exactly zero is due to noise from stochastic optimization.)

Our explanation of this property is that the SPoSE objective encourages multiple similar dimensions to be merged. Let $X$ be the $p \times m$ embedding matrix, and write $X_i$ for $i$th row of $X$. Suppose that the SPoSE model is well-specified and that there exist two highly similar dimensions in the true object space, $X_1$ and $X_2$, in the sense that $\Delta = X_1 - X_2$ is very small. Since the true similarity $S$ is defined as

$$S = X_1 X_1^T + \cdots + X_p X_p^T,$$

the contribution of these two dimensions to the true similarity matrix is $X_1 X_1^T + X_2 X_2^T$.

Rather than find these two dimensions from the data, SPoSE may find a single dimension

$$\hat{X}_1 \approx \frac{\sqrt{2}}{2}(X_1 + X_2).$$

This is because $\hat{X}_1$ closely approximates the contribution of $X_1$ and $X_2$ to the true similarity matrix $S$,

$$\hat{X}_1 \hat{X}_1^T = X_1 X_1^T + X_2 X_2^T - \frac{1}{2}\Delta\Delta^T \approx X_1 X_1^T + X_2 X_2^T.$$

But the L1 norm of $\hat{X}_1$ is a smaller by the factor $\sqrt{2}/2$ than the combined L1 norms of $X_1$ and $X_2$. Therefore, if the approximation error to the empirical log-likelihood from using $\hat{X}_1$ is small relative to the L1 penalty $\lambda(||X_1||_1 + ||X_2||_1)$, SPoSE will combine the two dimensions into one dimension. Note that the choice of L1 regularization is critical for this property: under the squared L2 norm, $\hat{X}_1$ incurs approximately the same penalty as $X_1$ and $X_2$ combined so that there is no incentive to merge.

This property has disadvantages and practical advantages. A possible disadvantage is that multiple dimensions in the data may be agglomerated if they are too similar. However, depending on the application, this may be an advantage rather than a disadvantage since a few agglomerated dimensions may be more easy to interpret than many seemingly redundant dimensions. The disadvantage is also not as much of a disadvantage when one considers that dimensions that are too similar may be impossible to disentangle based on empirical data anyways, so SPoSE does not fare worse in this regard than other methods.

This property of SPoSE allows for automatic dimensionality selection and also increases the reproducibility of the embedding.

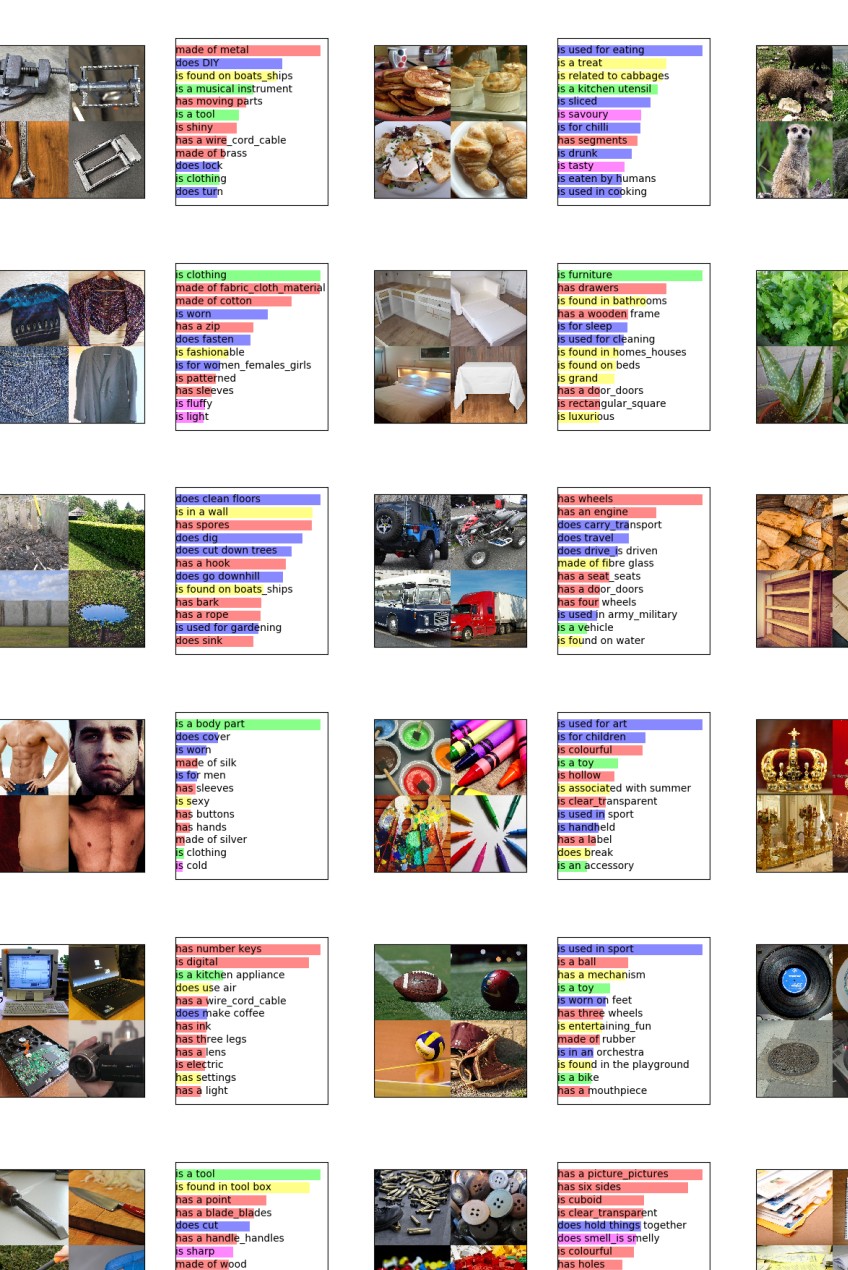

Figure 4: SPoSE features 1-21 (out of 49) explained by CSLB labels via NNLS. Top 4 concepts for each of four selected SPoSE feature are shown beside 12 CSLB features with the largest weights for predicting that SPoSE feature. Size of bars indicates relative weight, while color indicates feature type. Green: taxonomic; Blue: functional; Yellow: encyclopedic; Red: visual perceptual; Violet: non-visual perceptual.

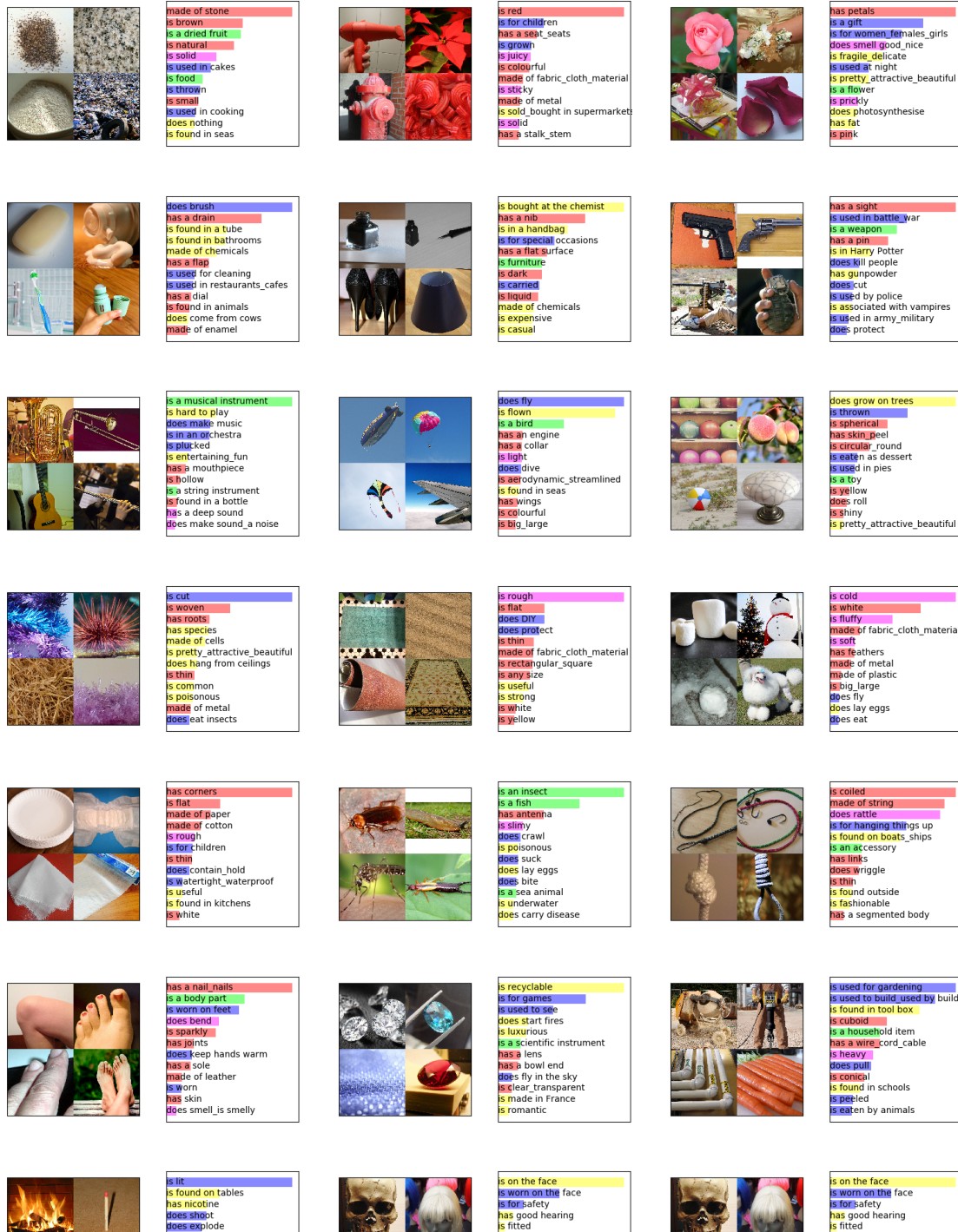

Figure 5: SPoSE features 22-42 (out of 49) explained by CSLB labels via NNLS. Top 4 concepts for each of four selected SPoSE feature are shown beside 12 CSLB features with the largest weights for predicting that SPoSE feature. Size of bars indicates relative weight, while color indicates feature type. Green: taxonomic; Blue: functional; Yellow: encyclopedic; Red: visual perceptual; Violet: non-visual perceptual.

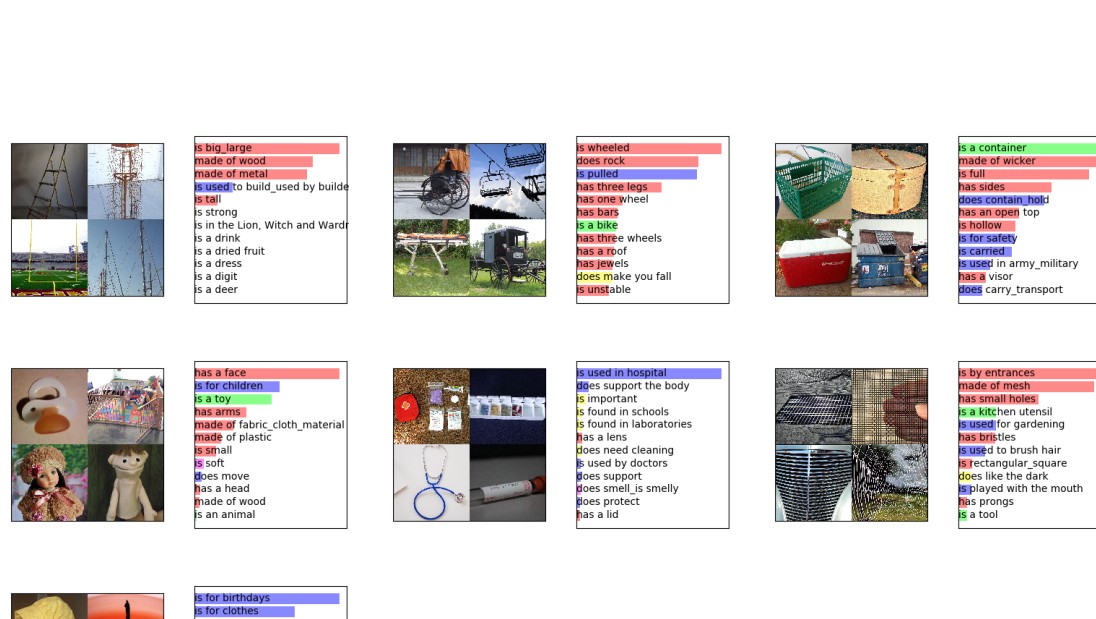

Figure 6: SPoSE features 43-49 (out of 49) explained by CSLB labels via NNLS. Top 4 concepts for each of four selected SPoSE feature are shown beside 12 CSLB features with the largest weights for predicting that SPoSE feature. Size of bars indicates relative weight, while color indicates feature type. Green: taxonomic; Blue: functional; Yellow: encyclopedic; Red: visual perceptual; Violet: non-visual perceptual.

