# OpenReview forum: "Revealing interpretable object representations from human behavior"
_ICLR.cc/2019/Conference_

### Official Review · AnonReviewer1 · 2018-11-02
**Behavioral experiment on human representations - Manuscript has improved -revison**

**Rating:** 5
**Confidence:** 4

**Review:**

This is a paper that communicates a large scale experiment on human object/semantic representations and a model of such representations.   The experiment could have been more carefully controlled (and described in the paper) and the modeling work is inconclusive.

Quality,
The experiment design is conventional, based on rating pair-wise similarity among triplets. Compared to earlier experiments, this data has more objects and more triplets.  Additional control experiments on smaller subsets have been carried out to further address hypotheses.  The description of the experiment could have been more careful: What are the precise instructions, how are the object/images presented (it is well known that relative positions, asymmetry, etc can play an important role), are there any temporal/learning effects (how clear is the task to the workers?).
The modeling work is basic and contains a number of steps that have unknown influence on the final outcome. For example model dimension: Is you claim that "D=49" is a law of human nature?  Model predictive performance seems excellent, that is interesting! But we do not know how robust this is to the many heuristics

Clarity,
The presentation of the inference process is clear. Not so clear what the uncertainties are

Originality
Limited. Mainly related to scale. But the data quality is unclear. The modeling approach involves a number of untested heuristics (non-negative, exponentiation etc).

Significance
Mostly related to the data.  I did not understand if it is planned to release the data.

Pros and cons

+Large scale experiment
+simple model, seem to have good accuracy

-experiment needs more careful description
-too many heuristics in model and inference, unclear how general the conclusions are

Other comments:
References have many issues

The authors have done a good job in the revision and have clarified points that were unclear in the first version.
I have remaining reservations on significance, but move rating up a notch to reflect the extensive improvements and  the authors' confirmation that they will release the data.

---

> ### Author Response · Authors · 2018-11-20
> **Response to Reviewer 1 - Part 3**
>
>
> > The modeling approach involves a number of untested heuristics (non-negative, exponentiation etc).
>
> We agree with the reviewer and provide more justification for our modeling choices below, as well in the updated manuscript. Note that, in the manuscript we already mention an analysis finding that embeddings based on the dot product and the Euclidean distance - both very common measures of psychological similarity / distance - are very similar in performance and interpretability.
>
> The use of Luce's choice rule with exponential weights is a very common approach for arbitrating between probabilistic choices, not only in supervised machine learning, but also in game theory and reinforcement learning (e.g. Sutton & Barto, 2001), and there is strong empirical support for its use in humans (e.g. Daw et al., 2006 - Nature). In the context of behavioral similarity, there is strong evidence that the probability of choice is exponentially related to psychological distance, as reviewed in Xu et al. (2011).
>
> The sparsity constraint is a reasonable assumption given that, without sparsity, all concepts would carry all dimensions, which would be in contrast to what is found empirically for feature norms of real-world objects that turn out to be sparse (see McRae et al. 2005). Note that if a non-sparse model was best for predicting the data, cross-validation would have revealed a lambda very close to 0 (our lambda was 0.0080, which was much larger than the grid spacing 0.0001, and as mentioned above, a penalty of 0.0270 or higher would result in all dimensions being shrunk to 0).
>
> Since our aim was to obtain interpretable embeddings, we used the constraint of non-negativity, which in the word embedding literature has been found to improve interpretability (Murphy et al. 2012). In response to the reviewer comment, we ran a similar analysis without non-negativity and sparsity constraints which, as expected, led to comparable performance (0.6453) and 18 dimensions which turned out to be much less interpretable. If “true” dimensions turn out to be signed, we expect SPoSE dimensions to either be shifted to the positive range or split into positive and negative parts. Importantly, this transformation does not affect their interpretability.
>
> **Updated / new text**
> “We assume that each feature/dimension x_if in the vector x_i is real and non-negative, so as to make it interpretable as the *degree* to which the aspect of meaning it represents is present and influences subject behavior (Murphy et al. 2012).. Further, in accordance with empirical findings, we expect features/dimensions to be sparse (McRae et al. 2005), which motivated us add a sparsity parameter to our model.”
>
> **References**
> Richard S. Sutton and Andrew G. Barto. Reinforcement learning: An introduction. MIT press, 2018.
>
> Nathaniel D. Daw, John P. O'Doherty, Peter Dayan, Ben Seymour, and Raymond J. Dolan. "Cortical substrates for exploratory decisions in humans." Nature 441(7095):876-879, 2006.
>
>
> > I did not understand if it is planned to release the data.
>
> Please note that under 2.1. (The Odd-One-Out Dataset), it reads "We plan to collect additional triplets and release all data by the end of the study."
>
>
> > References have many issues
>
> Thank you for bringing this to our attention. We have standardized the formatting of our references.

---

> ### Author Response · Authors · 2018-11-20
> **Response to Reviewer 1 - Part 2**
>
>
> > The modeling work is basic and contains a number of steps that have unknown influence on the final outcome. For example model dimension: Is you claim that "D=49" is a law of human nature?
>
> We do not attempt to recover the "true" dimensionality of the embedding, but rather a “useful” embedding to explain the data and provide interpretability. We can conclude from our results that there exist at least 49 useful embedding dimensions; it is possible that, after collecting more data, we might find more dimensions. As detailed in 2.2. under “Parameter fitting”, we fit a model with a large initial number of dimensions (in this case, 90), but the L1 shrinkage (with cross-validated lambda) naturally causes many of dimensions to have a maximum weight close to zero. We set a threshold to eliminate dimensions below a certain average weight, which results in an embedding with a much smaller number of dimensions. In our current dataset, this procedure resulted in picking an L1 penalty equal to 0.0080 (a penalty of 0.0270 or higher would result in all dimensions being shrunk to 0). We deleted all dimensions with average weight less than 0.02, resulting in the 49 dimensions presented.
>
>
> > The presentation of the inference process is clear. Not so clear what the uncertainties are
>
> For any triplet presented to a participant, there are two possible sources of randomness. One is variation in the decision-making process of the participant. The choice of the participant may be affected by biases such as ordering of the objects, and by uncontrolled factors such as their physiological state. A second source of variation is differences in the decision-making process between participants. Participants with different personalities and worldviews may evaluate the similarity of objects in different ways. Our model accounts for both sources of uncertainty by representing the choice for a given triplet for a random participant as a draw from a multinomial distribution, with probabilities given by equation 2, page 3.
>
>
> > But the data quality is unclear.
>
> See above comments clarifying the quality of the data. We have also added a description to the text specifying more precisely what the exclusion criteria were and how many triplets were excluded.
>
> **Updated / new text**
> “The dataset in this paper contains all of the data acquired to date, comprising judgments on 1,450,119 randomly selected triplets, roughly 0.13% out of all possible triplets using the 1,854 concepts. These were the triplets remaining after excluding AMT participants that showed evidence of inappropriate subject behavior, namely if responses were unusually fast, exhibited systematic response bias, or were empty (137,281 triplets or 8.65%).”

---

> ### Author Response · Authors · 2018-11-20
> **Response to Reviewer 1 - Part 1**
>
> We thank the reviewer for their assessment of our work. To address their criticisms, below and in the updated text we more carefully describe the experiment. In addition, we better justify the heuristics by being more explicit about how they are based on previous studies and empirical evidence, and we ran an additional analysis based on the reviewer’s comment on our modeling choices. Note that while the scale of our experimental data - as pointed out by the reviewer - strongly contributes to the significance of our work, we are not aware of comparable work that has revealed interpretable dimensions underlying human behavior from similarity ratings. It is, in fact, the *combination* of scale and the use of our sparse non-negative model that makes this work unique. As mentioned in the manuscript, all data will be released at the end of the study. We hope that our responses will provide the reviewer with the information sufficient to raise their rating above the acceptance threshold.
>
>
> > What are the precise instructions, how are the object/images presented (it is well known that relative positions, asymmetry, etc can play an important role), are there any temporal/learning effects (how clear is the task to the workers?).
>
> The reviewer raises a number of points that we did not address at length in the paper due to space constraints and which, in our opinion, were not critical to reproducing the results. However, we fully agree that it is important to add further justification, which we will do in this review and in the manuscript, as space permits. Please note that we also tried to better highlight some of the details (e.g. the release of the data) which were already present in the manuscript.
>
> The precise instructions to the workers were:
> “In each round, you will see three pictures each showing an object or "thing". Two of them will be more similar to each other. Your job is to select the *odd-one out* by clicking on it. Sometimes the decision is very difficult. Base your decision _only_ on the most prominent object or "thing" in an image.“
> Note that these instructions were intentionally left rather open, so as to allow workers to decide according to whatever criteria were most salient to them for carrying out the task.
>
> Regarding the issues mentioned by the reviewer (effects of temporal learning, effects of position, symmetry, etc.), we empirically tested the validity of the method in a separate study before acquiring the large-scale dataset. This study is extensive in itself and will thus be published separately. In it, we acquired similarity ratings for a set of 48 objects and a separate set of 92 objects using the triplet odd-one-out task. We then compared two other common similarity tasks (pairwise similarity and object arrangement) to the triplet odd-one-out task and related them to both synset embeddings and deep convolutional neural networks as well as human brain data (functional MRI and magnetoencephalography). The triplet odd-one-out task was highly correlated with the other two similarity tasks, overall, and performed equally well or better than those tasks in predicting embeddings and human brain data. This demonstrates that, all issues raised by the reviewer aside, the triplet odd-one-out task is as good or better than two common state-of-the-art alternatives for human similarity judgments.
>
> Note that any preference of position or sequence effects would only affect the variance of the estimates, not the bias. Most workers only carried out very few trials, weakening the possible contribution of learning effects to bias. If there was any strong bias present in the data, we believe that the model could not have performed as close to optimal as it did. However, we agree with the reviewer that, in the future, it would be interesting and important to investigate possible learning effects and, specifically, individual differences in how humans use those dimensions. This is, however, beyond the scope of the present work.

---

### Official Review · AnonReviewer2 · 2018-11-03
**Interesting paper**

**Rating:** 7
**Confidence:** 4

**Review:**

This is an interesting paper with a new approach to learn a sparse, positive (and hence interpretable) semantic space that maximizes human similarity judgements, by training to specifically maximize the prediction of human similarity judgements. The authors have collected the dataset themselves and have rating of sets of 3 objects from 1854 unique objects. They end up with a space (SPoSE) with relatively low dimensionality with respect to usual word embeddings (49 dimension) but perhaps not surprising when considering the small size of the words to embed. The authors run a set of experiment to show the usefulness of SPoSE. The most interesting one is the prediction of its dimensions by the CSLB features, which reveals a nice clustering in the different SPoSE dimensions. Perhaps the results would be a little more convincing if additional common word embeddings were also tested.

Due to the different objects used in the different datasets, some of the experiments have a smaller set of words. A good extension of this work would be to combine a text-derived embedding  or the synsets to interpolate the SPoSE dimensions for missing words in the original set. Or perhaps the object similarity ratings could be used in a semi-supervised setting to inform the learning of a co-occurence word embedding. This will allow the model to better describe a larger set of words. Another possible extension is to test this larger set of words on a non-behavioral NLP task to show possible improvements that the behavioral data and the interpretable space give.

---

> ### Author Response · Authors · 2018-11-20
> **Response to Reviewer 2**
>
> We thank the reviewer for their positive evaluation of our study. We agree that the prediction from CSLB features is particularly interesting, and we are currently working on improving this further by interpolating to other objects in a semi-supervised manner (similar to what was proposed by the reviewer). We also strongly agree that testing additional embeddings would be very interesting! For the present work, we focused on synset embeddings because they represent a closer match to the meaning of each individual object than word embeddings would and provide a one-to-one match for the meanings. For example, our list contains four different meanings for the object named by the word “baton”, referring to (1) an item in relay races, (2) in twirling, (3) a weapon used by police, and (4) an item used by a musical conductor. Due to the novelty of this line of research, to our knowledge there are no other synset embeddings available than the ones we used, and we included both a 50d dense and a 300d dense version. In addition, we would have liked to include sparse positive synset embeddings as a reference, however those are currently not available; for that reason, we included NNSE word embeddings instead. In the future, we would like to add sparse positive synset embeddings and test their interpretability relative to our similarity embedding. We hope this will underline the unique contribution of a behavior-based similarity embedding presented here.
>
> In addition, we would like to thank the reviewer for their idea on how to extend the embedding. Indeed, we are currently working on predicting similarities for other concepts and images from pretrained synset vectors and activations in deep convolutional neural networks. However, this effort is still in its early stages and beyond the scope of the present work.

---

### Official Review · AnonReviewer3 · 2018-11-03
**Interesting well done paper**

**Rating:** 7
**Confidence:** 4

**Review:**

Following the suggested rubric:
1. Briefly establish your personal expertise in the field of the paper.
2. Concisely summarize the contributions of the paper.
3. Evaluate the quality and composition of the work.
4. Place the work in context of prior work, and evaluate this work's novelty.
5. Provide critique of each theorem or experiment that is relevant to your judgment of the paper's novelty and quality.
6. Provide a summary judgment if the work is significant and of interest to the community.

1.  I work at the intersection of machine learning and biological vision
and have worked on modeling word representations.

2. This paper develops a new representation system for object
representations from training on data collected from odd-one-out human
judgements of images.  The vector representation for objects is
designed to be sparse and low dimensional (and ends up being about
49D).  Similarity is measured by dot products in the space and
probabilities of which pair of items will be paired are modeled as the
exponential of the similarity.


3,5  The resulting embedding	does a good job	of predicting human similarity
judgements and seems to cover similar features to those named by
humans.  They also explain typicality judgements and cluster semantic
categories well.   The creation of the upper limit based on noise between
and within subjects was a nice addition.


4. Some relevant related work is discussed and this seems like a novel
and interesting contribution.  The authors might also want to compare
to similar work that looked at similarities among triplets (Similarity
Comparisons for Interactive Fine-Grained Categorization
http://ttic.uchicago.edu/~smaji/papers/similarity-cvpr14.pdf;
Conditional Similarity Networks https://arxiv.org/abs/1603.07810 ).


6. While this paper is not especially surprising or ground breaking, the
number and quality of the comparisons make it a worthwhile
contribution and the resulting embeddings are worth further exploration
and could be very useful for future research.

---

> ### Author Response · Authors · 2018-11-20
> **Response to Reviewer 3**
>
> We thank the reviewer for their positive evaluation of our work. In response to their suggestion under Point 4, we have added the suggested references to sections 2.3 (Related Work) and 4 (Discussion and Conclusion)
>
> New text:
> “Similarly, Wah et al. (2014) show a series of adaptive displays for an anchor c_i, where the subject must partition the queries c_j, c_l, … into a set of similar and a set of dissimilar queries. In contrast to our work, the aforementioned studies did not use sparsity or positivity constraints, nor did they intend to evaluate the interpretability of the embedding.”
>
> New text:
> “Yet another possible extension is to consider different types of similarity judgments (Veit et al. 2017), e.g. resulting from asking subjects to group objects based on a specific attribute (size, color, etc.).”

---

### Author Response · Authors · 2018-11-20
**Changes to Manuscript in Response to Reviewer Comments**

We would like to thank all reviewers for their time and their thoughtful input. We did our best to address all of their comments in the individual responses below, which contain details not added to the paper itself due to space constraints. Based on the comments, we made the following changes to the paper:

• We more carefully describe the experiment / clarify the quality of the data in section 2.1.
• We clarify the justification for the sparsity and non-negativity properties of our model in section 2.2.
• We added references suggested by the reviewers.
• To make room for these additions, we shortened the introduction (“tomato example”), and compressed the text in a few places where it could be done without losing clarity.
• We fixed the formatting of some references and fixed the template from 2018 to 2019.

---

### Meta-Review · Area_Chair1 · 2018-12-15
**Reviewer consensus is accept**

**Confidence:** 4
**Recommendation:** Accept (Poster)

**Metareview:**

The reviewers viewed the work favorably, with only one reviewer providing a score slightly below acceptance. The authors thoroughly addressed the reviewer's original concerns, and they adjusted their score upwards afterwards. The low-rating reviewer remains skeptical of the significance of the work, but the other two reviewers make firm cases for the appeal of the work to the ICLR audience. In follow-up discussion after the author's responses were submitted and discussed, the low-rating reviewer did not make a clear case for rejecting the paper, and further, the higher-rating reviewers' arguments for the impact of the paper were convincing. Therefore, I recommend accepting this paper.